# Predictors of digital support services use by informal caregivers: a cross-sectional comparative survey

Alhassan Yosri Ibrahim Hassan  ,[1,2] Giovanni Lamura,[1] Mariët Hagedoorn[3]

¹Centre for Socio-Economic Research on Ageing, INRCA IRCCS, Italian National Institute of Health & Science on Ageing, Ancona, Italy
²Department of Economics and Social Sciences, Faculty of Economics "Giorgio Fuà", Marche Polytechnic University, Ancona, Italy
³University Medical Center Groningen, University of Groningen, Groningen, Netherlands

**Correspondence to**
Alhassan Yosri Ibrahim Hassan, Centre for Socio-Economic Research on Ageing, INRCA IRCCS, Italian National Institute of Health & Science on Ageing, Ancona, Italy;
hassanyousri@hotmail.com; a.hassan@inrca.it

## ABSTRACT

**Objectives** Digital support services may provide informal caregivers with remote access to information and training about care issues. However, there is limited specific data on how factors such as demographics, socioeconomic resources and the caregiving context may influence caregivers' use of digital support services. The aim of this study is to identify associations between informal caregiver's characteristics and the use of the internet to access digital support services in two countries: Italy and Sweden.

**Setting and participants** A sample of 663 respondents who have access to the internet participated in a cross-sectional survey by completing the online questionnaire. Respondents were recruited by the Italian National Institute of Health and Science on Ageing and the Swedish Family Care Competence Centre.

**Primary and secondary outcome measures** Logistic regression analyses were performed to assess predictors of caregivers' frequent use of the internet to access digital support services.

**Results** Educational attainment (OR 3.649, 95% CI 1.424 to 9.350, p=0.007), hours per week spent caring (OR 2.928, 95% CI 1.481 to 5.791, p=0.002), total household income (OR 0.378, 95% CI 0.149 to 0.957, p=0.040), care recipient relationship to the caregiver (OR 2.895, 95% CI 1.037 to 8.083, p=0.042) and gender of care recipient (OR 0.575, 95% CI 0.356 to 0.928, p=0.023) were significant predictors in the multivariate analysis for the Italian caregivers group. Hours per week spent caring (OR 2.401, 95% CI 1.105 to 5.218, p=0.027) and age of caregiver (OR 2.237, 95% CI 1.150 to 4.352, p=0.018) were significant predictors in the multivariate analysis for the Swedish caregivers group.

**Conclusions** Digital support services could be important tools to empower informal caregivers. When it comes to policy and practice in relation to caregivers, similarly to other broad vulnerable groups, there is no 'one-size-fits-all' approach, and it is therefore important to consider the specific characteristics and needs of both caregivers and care recipients.

## INTRODUCTION

Informal caregivers are individuals who provide care to ill, frail or disabled relatives, friends or others, without being trained or paid, in contrast to formal caregivers who offer professional services.[1] In Europe, 80%

**Strengths and limitations of this study**

► This study is an international comparative study investigating the important factors associated with the use of digital support services among informal caregivers.

► Multivariate logistic regression analyses enabled the effect of confounding factors to be controlled for and predictors of use of digital support services among informal caregivers to be identified.

► Given the cross-sectional design of our study, causal relationships cannot be established.

► The survey was conducted using the internet, and thus, our findings may not be generalizable to individuals who do not use the internet.

of all care is provided by informal caregivers who are often females, either providing care to a spouse, parent or parent-in-law, and a large share is provided by individuals who are older than standard retirement age.[2–4] Estimates on the economic value of unpaid informal care in the European Union (EU) Member States range from 50% to 90% of the overall costs of formal long-term care provision.[4] The available estimates of the number of informal caregivers ranges from 10% up to 25% of the total population in Europe.[5] The number of informal caregivers over 18 years of age who provide more than 20 hours per week of informal care to older adults and relatives with disability is estimated to be more than 70 million.[5] Informal caregivers provide the bulk of long-term care, including via contributions to both activities of daily living (personal care, feeding, dressing and grooming, emotional and social support, etc) as well as instrumental activities of daily living (transportation, care coordination, etc)

Caregiving may prove challenging and stressful for many informal caregivers. Caregivers often experience high levels of need for information and services. Available literature points to the importance of novel technology solutions as a promising

approach for empowering and supporting informal caregivers.[6–8] Digital support services for informal caregivers are services provided by any private or public organisation that address caregivers and/or care recipients' needs through technological devices that are integrated or not into a wider intervention programme.[9] Digital support services may provide informal caregivers with remote access to information and training about care and caring-related issues through websites, mobile applications and online training materials.[10] These solutions may contribute to a more positive caregiving experience and may help to strengthen informal caregivers' sense of social inclusion and belonging.[11] Digital support services also have macro-level benefits as these solutions may help in the integration of informal and formal care through better care coordination and a reduction in unnecessary hospitalisations and lengths of stay.[8–12] Consequently, the deployment of these solutions may generate savings and contribute to the sustainability of care systems.[8–12]

Considering the substantial information needs experienced by informal caregivers, the increased availability of digital support services for caregivers as well as the potential they offer, further understanding of caregivers' use of the internet to access digital support services is needed,[13 14] in order to determine whether factors such as demographics, socioeconomic resources and the caregiving context may influence caregivers' use of digital support services.[15] Previous literature on internet use for health information seeking showed that young age, good health status and higher education are associated with a more frequent use.[16–28] Prior studies also found that females were likely to seek health information on the internet more frequently than males.[16–28]

Italy and Sweden represent two European extremes with respect to several dimensions. These include: familistic/universalistic orientation of care system (Italy: family based, Sweden: universal); the level of overall digital skills (low in Italy: 42%, high in Sweden: 72%); and that of internet use for health information-seeking (low in Italy: 35%, high in Sweden: 62%).[29–32] The two countries share however also some similarities. Both Italy and Sweden are high-income countries and represent two of the oldest populations in Europe,[33 34] also because they report an almost similar, very high life expectancy at birth, estimated at 83 and 82 years for Italy and Sweden, respectively.[33 34] Estimates on the prevalence of informal care in Italy ranges from 14% up to 26% of the country's population.[35] In Sweden, it is estimated that 18% of the 18+ population provides informal care on a regular basis, corresponding to over 1.3 million people overall.[36]

In the literature, very few studies exclusively focus on caregivers' use of the internet to access support services. While informal caregivers have been identified as a population group which could benefit from the provision of digital support services, there is limited specific data on how factors such as demographics, socioeconomic resources and the caregiving context may influence caregivers' use of the internet to access digital support services. Mapping the sociodemographic and socioeconomic profiles of informal caregivers who do use and those who not use digital support services could help improve the quality of these services available to them. The aim of this study is, therefore, to identify associations between informal caregiver's characteristics and the use of the internet to access digital support services in two countries: Italy and Sweden. Exploring the experiences of informal caregivers in accessing digital support services in these two countries could inform future reforms of the healthcare system, and boost caregivers' access to information, services and support via new technologies in accordance to their needs. Moreover, since health promotion and patient empowerment via digital technologies are also on the European agenda,[37] exploring the commonalities and differences in informal caregivers' access to digital support services in these two countries, could contribute to provide recommendations useful for implementing the EU agenda on the transformation of the digital health and care agenda, while responding to caregivers' needs in each country.

## METHODS
### Study design
This online survey study used a cross-sectional design to identify associations between informal caregiver's characteristics and the use of internet to access digital support services in two countries: Italy and Sweden. The data presented here, aimed at evaluating technology based support services for informal caregivers, were collected through the support of a partnership of different stakeholders belonging to the Eurocarers' network (European Association Working for Carers). They represent national-level caregiver organisations in mostly EU Member States as well as research centres working on these topics, such as the Centre for Socio-Economic Research on Ageing of Italy's National Institute of Health and Science on Ageing, the Swedish Family Care Competence Centre and the Department of Economics and Social Sciences of Marche Polytechnic University (Italy).

### Survey administration
The sample was identified from the registries of the Italian National Institute of Health and Science on Ageing and the Swedish Family Care Competence Centre. The online survey link was disseminated from November 2020 to April 2021 through the different communication channels, that is, mailing lists and official websites, of the Italian National Institute of Health and Science on Ageing and the Swedish Family Care Competence Centre. Study participants were included provided they were:
► Informal caregivers of dependent adult individuals living at home with access to the internet.
► Eighteen years old and above.
► And either resident in Italy and able to understand Italian (for participants answering the Italian version of the questionnaire), or resident in Sweden and able

to understand Swedish (for participants answering the Swedish version of the questionnaire).

Exclusion criteria were as follows:

► Informal caregivers of paediatric patients.
► Professional or paid caregivers.
► People with medical comorbidities that prevent them from completing the questionnaire (eg, cognitive impairments).

The study sample included respondents who classified themselves as informal caregivers based on the survey question: 'Do you provide unpaid care at home to an adult relative, neighbour or friend to help them take care of themselves?'. Participants were asked to answer this question with 'yes' or 'no,' and if they answered 'yes', then they were asked to continue with the questionnaire. A unique identification number was provided to each participant and stored together with the survey results, in order to eliminate duplicate entries. The participants were given the option to save their responses and return to complete the survey, or they could edit or clear the replies and initiate the survey another time. All no respondents received email reminders. The response rate is estimated to be 31%. Data were recorded in the system using a password-protected data extraction form.

## Variables and measurement

Guided by Wilson's model of information-seeking behaviour,[38] the previous survey on services for supporting family carers of older dependent people in Europe 'EUROFAMCARE',[39] and empirical evidence in the literature,[16–28] this study included the following sets of independent variables: caregiver's demographics; caregiver's socioeconomic resources and caregiving context. The dependent variable in this study is informal caregivers' frequent use of the internet to access digital support services. In the survey, caregivers were asked to report how frequently they were using the internet to access digital support services. Those using the internet at least several times per month to access digital support services were classified as 'frequent users', while those accessing it less often were classified as 'infrequent users'. Three demographic measures were included: caregiver's age, caregiver's gender and caregiver's health status. Ages were measured in chronological years and grouped into three categories: 18–39, 40–59 and 60 or older. Gender was measured nominally and was grouped into male and female. Caregiver's health status was grouped into poor, fair and good. Measures of social and economic circumstances were the caregiver's educational attainment and their total household income. Educational attainment was grouped into primary, secondary, bachelor's degree and higher than bachelor's degree. Income was assessed by asking the caregiver about their 'monthly household net income from all sources'. In order to distribute the income by different income groups and enhance the cross-national comparability of results between the two countries involved in this study, Italy and Sweden, we referred to the official figures of the national median

equivalised disposable annual income from the European Commission's European statistical system 'Eurostat'.[32] We used these official figures in classifying the participants into three groups of household net income in each of these two countries:

1. Lower-income group: income is less than below 50% of the national median equivalised disposable annual income. This is equivalent to an income lower than 5802 Euro in the case of Italy and an income lower than 9356 Euro in the case of Sweden.
2. Middle-income group: income is between below 50% of the national median equivalised disposable annual income and above 60% of the national median equivalised disposable annual income. This is equivalent to an income between €5802 and €19 658 in the case of Italy and an income between €9356 and €26 826 in the case of Sweden.
3. Upper-income group: income is higher than €19 658 in the case of Italy and higher than €26 826 in Sweden.

Caregiving context was assessed using the following variables: reported number of weekly hours of care provided to the care recipient; reported number of years spent providing care; age and gender of the care recipient; relationship between the care recipient and the caregiver; and the level of dependency of the care recipient. Responses concerning the average number of weekly hours of caregiving have been grouped into four categories: (1) 10 hours or less, (2) 11–20 hours, (3) 21–40 hours and (4) more than 40 hours. Care duration was measured on the basis of the caregiver's reported length of care provision to the care recipient (in number of years), and respondents were classified into two groups: those caring for 2 years or less; and those caring for a longer time. The age of the care recipient was reported according to two groups: 60 years or less and more than 60 years. The gender of care recipients was grouped into male and female. Caregivers were requested to provide information about the person whom they care for, in order to assess the relationship with the care recipient (eg, parents/parents-in-law, spouse/partner, friend/neighbour, child or other relative. The level of dependency of the care recipient on the caregiver was clustered in two groups: high dependency and low dependency.

## Data analysis

The data analysis was conducted in three stages. It began with univariate analyses including percentages to describe the characteristics of this sample of caregivers. At the second stage, the relationship between the outcome variable and the independent variables was examined using Pearson's $\chi^2$ test with Yates' continuity correction. Differences between groups were considered significant at the 5% level (p≤0.05). Contingency tables have been assessed, before proceeding to logistic regression, to ensure there were no cells with expected frequencies of fewer than 5 to prevent biased estimates.[40] At the last stage, logistic regression analysis was used to establish the ability of each variable to predict caregivers' frequent use of the internet

to access digital support services while controlling the effects of other variables. Variables identified as statistically significant in the bivariate analysis were entered into logistic regression analysis for each measure of use of the internet to access digital support services.

The logistic regression analyses produced ORs with 95% CIs to identify predictors of each measure. Results are reported in ORs, which can be interpreted as the ratio of the probability that caregivers with a particular characteristic (eg, male gender) will use the internet frequently to access digital support services, over the probability they will use the internet frequently to access digital support services, had they not this characteristic. ORs that are higher than 1 indicate a positive association between a given variable and using the internet frequently to access digital support services, while an OR lower than 1 indicates a negative association. Statistical analyses were performed using SPSS software V.28.0 (IBM).

Informal caregivers expressing interest in participating in the study were informed about the aim of the study, the expected time to complete the questionnaire, and that data would be stored by the Centre for Socio-Economic Research on Ageing of the Italian National Institute of Health and Science on Ageing. The technical functionality of the online questionnaire had been tested before fielding the questionnaire. The estimate time for survey completion was 10–15 minutes. Informed consent was obtained from all participants. No personal information about the participants such as their name or their IP address were collected. All the responses were anonymous.

### Patient and public involvement
Patients or the public were not involved in the design, or conduct, or reporting, or dissemination plans of our research.

## RESULTS
### Sample description
A total of 663 informal caregivers, 410 from Italy and 253 from Sweden, participated in the survey by completing the online questionnaire. Table 1 presents the overall characteristics of the sample. Females represented a majority of respondents in the Italian group. The median age of caregivers was 54 years while the median age of care recipients was 73 years. Most Italian participants were providing care to a parent (n=163, 39.8%), to a female care recipient (n=223, 54.4%), spent more than 40 hours per week providing care (n=170, 41.5%) and had completed secondary school or lower (n=254, 62%). Nearly half of the participants (n=196, 47.8%) had an annual household income of less than 19658 Euro. The big majority of caregivers in the Italian sample (n=342, 83.4%) reported a fair or poor health status, provided care to a highly dependent care recipient (n=329, 80.2%) and had been providing care for more than 2 years (n=287, 70%) (table 1).

**Table 1** Characteristics of the sample (total sample N=663)

| Variables | Italian sample n=410 n (%) | Swedish sample n=253 n (%) |
|---|---|---|
| Gender | | |
| Male | 93 (22.7) | 57 (22.5) |
| Female | 317 (77.3) | 196 (77.5) |
| Age | | |
| Median | 54 | 65 |
| 18–39 | 48 (11.7) | 17 (6.7) |
| 40–59 | 241 (58.8) | 77 (30.4) |
| More than 60 | 121 (29.5) | 159 (62.8) |
| Health status | | |
| Good | 68 (16.6) | 43 (17.0) |
| Fair | 171 (41.7) | 155 (61.3) |
| Poor | 171 (41.7) | 55 (21.7) |
| Education | | |
| Primary | 29 (7.1) | 35 (13.8) |
| Secondary | 225 (54.9) | 114 (45.1) |
| Bachelor | 114 (27.8) | 61 (24.1) |
| Higher than bachelor's degree | 42 (10.2) | 43 (17.0) |
| Income | | |
| Lower | 39 (9.5) | 17 (6.7) |
| Middle | 157 (38.3) | 92 (36.4) |
| Upper | 214 (52.2) | 144 (56.9) |
| Care recipient relationship to caregiver | | |
| Parents (in law) | 163 (39.8) | 63 (24.9) |
| Spouse/partner | 64 (15.6) | 97 (38.3) |
| Child | 105 (25.6) | 48 (19.0) |
| Friend/neighbour | 30 (7.3) | 26 (10.3) |
| Other | 48 (11.7) | 19 (7.5) |
| Gender of care recipient | | |
| Male | 187 (45.6) | 136 (53.8) |
| Female | 223 (54.4) | 117 (46.2) |
| Age of care recipient | | |
| Median | 73 | 75 |
| 60 or younger | 160 (39.0) | 73 (28.9) |
| More than 60 | 250 (61.0) | 180 (71.1) |
| Level of dependency of the care recipient | | |
| High dependency | 329 (80.2) | 139 (54.9) |
| Low dependency | 81 (19.8) | 114 (45.1) |
| Hours spend caring each week | | |
| 10 hours or less | 115 (28.0) | 112 (44.3) |

Continued

**Table 1** Continued

| Variables | Italian sample n=410 n (%) | Swedish sample n=253 n (%) |
|---|---|---|
| 11–20 hours | 68 (16.6) | 62 (24.5) |
| 21–40 hours | 57 (13.9) | 30 (11.9) |
| More than 40 hours | 170 (41.5) | 49 (19.4) |
| Number of years providing care | | |
| 2 years or less | 123 (30.0) | 111 (43.9) |
| More than 2 years | 287 (70.0) | 142 (56.1) |

When compared with their Italian counterparts, both Swedish participants and their care recipients had a higher median age of 65 and 75 years, respectively. Females made up a majority of participants in the Swedish sample. Most of the Swedish respondents reported providing care to a spouse/partner (n=97, 38.3%), a male care recipient (n=136, 53.8%), spent less than 10 hours per week providing care (n=112, 44.3%) and had completed a secondary school or lower (n=149, 58.9%). Nearly half of the participants in the Swedish group (n=109, 43.1%) had annual household incomes less than 26826 Euro. The majority of the caregivers in the Swedish sample (n=210, 83%) had a fair or poor health status, were caring for a highly dependent care recipient (n=139, 54.9%) and had been providing care for more than 2 years (n=142, 56.1%) (table 1).

### Factors associated with caregivers' frequent use of the internet to access digital support services

Table 2 shows the factors associated with caregivers' frequent use of the internet to access digital support services in the bivariate analysis for each of the two countries of the study. In the Italian group, two-thirds of the respondents reported using the internet at least several times per month to access digital support services. At the bivariate level, this was associated with two demographic variables, caregiver's age and health status, and two socio-economic measures, caregiver's educational attainment and total household income. Five measures of caregiving context—care recipient relationship to the caregiver, gender of care recipient, age of care recipient, hours per week spent caring and the level of dependency of the care recipient—were also linked to the frequent use of the internet to access digital support services.

In the Swedish sample, 54.2% of the participants reported using the internet at least several times per month to access digital support services. In the bivariate analysis, caregiver's age was significantly associated with the frequent use of the internet to access digital support services. Three measures of caregiving context were also linked with the frequent use of the internet to access digital support services: care recipient relationship to the caregiver, age of care recipient and the number

of hours spent caring each week. None of the measures of socioeconomic resources was significantly associated with frequent use of the internet to access digital support services in the Swedish sample.

### Predictors of caregivers' frequent use of the internet to access digital support services

Table 3 summarizes the results of the logistic regression analysis predicting caregivers' frequent use of the internet to access digital support services. For the Italian sample, nine variables significantly associated with a frequent use of the internet to access digital support services in the bivariate analysis were entered into logistic regression analysis to identify which were predictive: caregiver's age, health status, educational attainment, total household income, care recipient relationship to the caregiver, gender of care recipient, age of care recipient, number of weekly hours of care and the level of dependency of the care recipient. The multivariate analysis indicated that educational attainment, number of weekly hours of care, total household income, care recipient relationship to the caregiver and gender of care recipient remained significant predictors. The strongest predictor was the educational attainment of the caregivers. Informal caregivers who completed education equivalent to a Bachelor's degree level had 3.649 times the odds of using the internet at least several times per month to access digital support services compared with those who completed a primary education (p=0.007, 95% CI 1.424 to 9.350). Caregivers who spend more than 40 hours per week providing care were almost three times more likely to be frequent users of the internet to access digital support services in comparison with those who spend 10 hours or less per week providing care. The odds of frequent use of the internet to access digital support services were 2.646 times higher for caregivers belonging to the lower household income group compared with caregivers belonging to the upper household income group (p=0.040, 95% CI 0.149 to 0.957). Regarding the relationship between the caregiver and care recipient, the caregivers of a child had 2.895 times the odds of using the internet at least several times per month to access digital support services compared with those who provide care to another relative (p=0.042, 95% CI 1.037 to 8.083). The odds of frequently accessing digital support services were 1.739 times higher for caregivers who provide care to a male care recipient compared with those providing care to a female care recipient (p=0.023, 95% CI 0.356 to 0.928).

The logistic regression analysis to predict the frequent use of the internet to access digital support services among Swedish participants consisted of the four statistically significant factors identified in the bivariate analysis: caregiver's age, care recipient relationship to the caregiver, age of care recipient and the number of weekly hours of care (table 3). The number of weekly hours of care remained a significant predictor in the multivariate analysis for the Swedish sample and was the strongest predictor. Swedish respondents who spend more than

**Table 2** Factors associated with caregivers' frequent use of the internet to access digital support services in the bivariate analysis

| Variables | Using the internet at least several times per month to access digital support services | | | |
| | Italian sample n=410 | | Swedish sample n=253 | |
| | n (%) | P value* | n (%) | P value |
|---|---|---|---|---|
| All respondents | 274 (66.8) | | 137 (54.2) | |
| Gender | | 0.123 | | 0.344 |
| Male | 56 (60.2)† | | 34 (59.6) | |
| Female | 218 (68.8) | | 103 (52.6) | |
| Age | | 0.01 | | 0.035 |
| 18–39 | 23 (47.9) | | 6 (35.3) | |
| 40–59 | 169 (70.1) | | 50 (64.9) | |
| More than 60 | 82 (67.8) | | 81 (50.9) | |
| Health status | | 0.042 | | 0.268 |
| Good | 37 (54.4) | | 35 (63.6) | |
| Fair | 115 (67.3) | | 79 (51.0) | |
| Poor | 122 (71.3) | | 23 (53.5) | |
| Education | | 0.008 | | 0.901 |
| Primary | 12 (41.4) | | 20 (57.1) | |
| Secondary | 161 (71.6) | | 60 (52.6) | |
| Bachelor's degree | 76 (66.7) | | 32 (52.5) | |
| Higher than bachelor's degree | 25 (59.5) | | 25 (58.1) | |
| Income | | 0.025 | | 0.736 |
| Lower | 32 (82.1) | | 10 (58.8) | |
| Middle | 110 (70.1) | | 47 (51.1) | |
| Upper | 132 (61.7) | | 80 (55.6) | |
| Care recipient relationship to caregiver | | <0.001 | | 0.014 |
| Parents (in law) | 95 (58.3) | | 23 (36.5) | |
| Spouse/partner | 49 (76.6) | | 55 (56.7) | |
| Child | 88 (83.8) | | 33 (68.8) | |
| Friend/neighbour | 16 (53.3) | | 15 (57.7) | |
| Other | 26 (54.2) | | 11 (57.9) | |
| Gender of care recipient | | <0.001 | | 0.732 |
| Male | 141 (75.4) | | 75 (55.1) | |
| Female | 133 (59.6) | | 62 (53.0) | |
| Age of care recipient | | 0.002 | | 0.037 |
| 60 or younger | 121 (75.6) | | 47 (64.4) | |
| More than 60 | 153 (61.2) | | 90 (50.0) | |
| Level of dependency of the care recipient | | 0.032 | | 0.853 |
| High dependency | 228 (69.3) | | 76 (54.7) | |
| Low dependency | 46 (56.8) | | 61 (53.5) | |
| Hours spend caring each week | | <0.001 | | 0.022 |
| 10 hours or less | 57 (49.6) | | 49 (43.8) | |
| 11–20 hours | 46 (67.6) | | 38 (61.3) | |
| 21–40 hours | 38 (66.7) | | 17 (56.7) | |
| More than 40 hours | 133 (78.2) | | 33 (67.3) | |

Continued

**Table 2** Continued

| | Using the internet at least several times per month to access digital support services | | | |
| | Italian sample n=410 | | Swedish sample n=253 | |
| Variables | n (%) | P value* | n (%) | P value |
| No of years providing care | | 0.464 | | 0.213 |
| 2 years or less | 79 (64.2) | | 65 (58.6) | |
| More than 2 years | 195 (67.9) | | 72 (50.7) | |

*Differences between groups were considered significant at the 5% level (p≤0.05).
†Male caregivers who are frequently using the internet as a % of the total number of male caregivers in the sample.

40 hours per week providing care were almost 2.5 times more likely to be frequent users of the internet to access digital support services as opposed to those who dedicate 10 hours or less per week to care provision (p=0.027, 95% CI 1.105 to 5.218). The age of the caregiver also remained a significant predictor in the multivariate analysis. Caregivers in the age group 40–59 years were 2.237 times more likely to use the internet at least several times per month to access digital support services in comparison with those of the age group 60+ years (p=0.018, 95% CI 1.150 to 4.352).

## DISCUSSION
### Principal findings
The purpose of this study was to identify important factors related to caregivers' use of the internet to access digital support services in Italy and Sweden. The findings suggest that a number of demographic, socioeconomic and caring circumstances are associated with the frequency of using the internet to access digital support services among caregivers in both countries. Multivariate regression analyses enabled the effect of confounding factors to be controlled for and predictors of use to be identified. In consistency with literature on the same topic in different countries,[16–28] our findings indicate that caregiver's age, health status, caregiver's educational attainment, total household income, care recipient relationship to the caregiver, gender of care recipient, age of care recipient, hours per week spent caring and the level of dependency of the care recipient are all associated with use.

The study shows that more than half of the caregivers in both countries frequently use the internet to access digital support services. While the use of the internet for health information has been somewhat less common in Southern European countries, in our study the Italian and the Swedish groups report an almost similar use of the internet to access digital support services. This may be related to the lower median age of the Italian sample compared with the Swedish one. Caregivers from Southern European countries with a family-based care system often lack support in terms of formal services and professional training from the government.[41–48] This shortcoming of support may increase their need for information and services. Digital support services may be an alternative support source that enables remote access to information and training about care and caring-related issues. Previous studies suggested that the use of the internet for health information in Southern European countries is increasing, and that caregivers from this region are showing an increased interest in accessing new technologies aiming to support them.[48–50]

In both countries, most of the caregivers who participated in the study were females, which is consistent with the results of previous works[25–28] and with the central role played by females in the provision of informal care.[2–4] In coherence with previous literature,[39 41–48] the majority of Italian participants in our study provided care to a parent (in law) and spent more than 40 weekly hours of care, compared with their Swedish counterparts who provided care to spouse/partner and spent less than 10 hours per week providing care. Previous research showed that care for someone in one's own household is more common in Southern European countries than in Northern countries. In Southern countries, caregivers are more likely to live with their care recipients who often are parents/in-laws.[39 41–48] In Northern countries, in-household care is mostly spouse care, as it is rare for old persons to live with anyone else than their spouse. Consequently, caregivers from Southern European countries spend more hours in caregiving compared with caregivers from Northern countries.

As it is to be expected given previous research on using the internet for general health information,[19–24] the digital divide may negatively affect caregivers' use of the internet to access digital support services.[51 52] The socioeconomic status of users seems to be a significant factor that increases the digital divide in Southern European countries.[53–55] This was apparent in our study, showing that the divide was more significant in the case of the Italian group compared with the Swedish one. While none of the measures of socio-economic resources was significantly associated with a frequent use of the internet to access digital support services in the Swedish group, the strongest predictor for the frequency of internet use in the Italian group was the caregiver's educational attainment. Previous research has shown that better-educated caregivers are more likely to be engaged in more frequent online activities.[25–28] Income was also a predictor for

**Table 3** Multivariate logistic regressions: caregivers' frequent use of the internet to access digital support services

| | Using the internet at least several times per month to access digital support services | | | | | |
| | Italian sample n=410 | | | Swedish sample n=253 | | |
| Variables | P value | OR | 95% CIs | P value | OR | 95% CIs |
|---|---|---|---|---|---|---|
| Age (in years) (Ref.: 60+) | | | | | | |
| 18–39 | 0.27 | 0.63 | 0.277 to 1.433 | 0.653 | 0.761 | 0.231 to 2.508 |
| 40–59 | 0.563 | 1.175 | 0.680 to 2.030 | 0.018 | 2.237 | 1.150 to 4.352 |
| Health status (Ref.: Good) | | | | -* | – | – |
| Fair | 0.703 | 1.105 | 0.661 to 1.850 | | | |
| Poor | 0.925 | 1.033 | 0.523 to 2.040 | | | |
| Education (Ref.: Primary) | | | | – | – | – |
| Secondary | 0.008 | 3.236 | 1.358 to 7.711 | | | |
| Bachelor | 0.007 | 3.649 | 1.424 to 9.350 | | | |
| Higher than bachelor's degree | 0.077 | 2.624 | 0.901 to 7.647 | | | |
| Income (Ref.: Lower) | | | | – | – | – |
| Middle | 0.17 | 0.514 | 0.198 to 1.331 | | | |
| Upper | 0.04 | 0.378 | 0.149 to 0.957 | | | |
| Care recipient relationship to caregiver (Ref.: Other) | | | | | | |
| Parents (in law) | 0.554 | 0.797 | 0.376 to 1.688 | 0.086 | 0.37 | 0.119 to 1.150 |
| Spouse/partner | 0.337 | 1.611 | 0.608 to 4.267 | 0.634 | 0.777 | 0.275 to 2.196 |
| Child | 0.042 | 2.895 | 1.037 to 8.083 | 0.911 | 1.075 | 0.302 to 3.828 |
| Friend/neighbour | 0.673 | 0.806 | 0.297 to 2.192 | 0.885 | 1.095 | 0.320 to 3.744 |
| Gender of care recipient (Ref.: male) | | | | – | – | – |
| Female | 0.023 | 0.575 | 0.356 to 0.928 | | | |
| Age of care recipient (Ref.: 60 or younger) | | | | | | |
| More than 60 | 0.211 | 1.616 | 0.762 to 3.424 | 0.92 | 1.046 | 0.436 to 2.511 |
| Level of dependency of the care recipient (Ref.: High dependency) | | | | | | |
| Low dependency | 0.738 | 1.111 | 0.599 to 2.062 | – | – | – |
| Hours spend caring each week (Ref.: 10 hours or less) | | | | | | |
| 11–20 hours | 0.021 | 2.241 | 1.127 to 4.459 | 0.085 | 1.822 | 0.921 to 3.602 |
| 21–40 hours | 0.103 | 1.908 | 0.878 to 4.144 | 0.311 | 1.568 | 0.656 to 3.748 |
| More than 40 hours | 0.002 | 2.928 | 1.481 to 5.791 | 0.027 | 2.401 | 1.105 to 5.218 |

*Only variables significantly associated with using the internet frequently to access digital support services in the bivariate analysis were entered into multivariate logistic regression analysis.

the frequency of internet use to access digital support services in the Italian group, with higher odds for caregivers belonging to the lower household income group. While literature suggests that general internet users in higher-income households are more likely than others to go online frequently,[56 57] previous studies on the internet use for health-related activities suggest that lower-income households may be more likely than others to go online for support activities.[16 58 59] One possible explanation is that those with higher incomes may have other means of

support, while those with lower incomes may turn to the internet as an alternative source of assistance.

The literature shows that age is a factor associated with internet use.[16–28] In the Swedish group of our study, age is a significant predictor of frequency of use. Age remained an important predictor of use when the effects of other demographics, socioeconomic factors and caring circumstances had been controlled for. This suggests that the relationship between age and use among Swedish caregivers cannot be entirely explained by increased financial

hardship in later life. Previous research suggests that use of the internet for health information is relatively constant by age, until age 65 when it begins to decline.[16]

Patterns of use among caregivers in both countries also seem to be shaped by the caring experience. The number of weekly hours of care was a significant predictor for the frequency of internet use by participants in both countries. Evidence from literature suggests that high-intensity caregivers report higher levels of information and service needs.[60] Given the availability and convenience of online sources, high-intensity caregivers may turn to the internet for digital support services.

## Limitations

Some limitations concerning this study need to be considered. The risk of the typical sampling bias should be mentioned as higher income and more educated caregivers are more likely to participate in research studies involving modern technologies, which was the case in our study. Furthermore, the sample size, especially of the Swedish sample, prevented us from carrying out more sophisticated statistical analyses. Moreover, not all of those who provide informal caregiving and assistance to others identify themselves as informal caregivers; consequently, we may have failed to capture the experiences of these underrepresented groups. We may have also failed to capture the concerns of the caregivers that may limit their ability in using digital resources. Although the most important variables identified from empirical evidence in the literature were included in the models, residual external variables may still have influenced our results. Conclusions drawn from this study results must be tempered by the fact that respondents were already possessing minimal digital skills that would enable them to access online services. It is possible that those who are not interested or involved with technology or those with limited digital access are less likely to respond to online surveys; consequently, the data collected online might be skewed and the sample might be less representative for the population. These issues might have influenced our findings and underline the need to interpret the findings from this study and other studies on caregivers with some caution when generalizing the findings.

## Recommendations and implications

Our results indicate that digital support services may enable remote access to information and training about care and caring-related issues. In this context, looking for information and support services online may be considered an attempt to close some knowledge gap. With the rapid technology advancement and increased access to the internet, more caregivers are expected to access these services.[61] This suggests that the interaction with informal caregiver by healthcare professionals and other parties with an interest in supporting them (eg, caregiver advocacy organisations) is an integral part of the value chain that supports both communication and coordination of services. Hence, these parties should all be more

engaged with developing digital support services targeted at informal caregivers, and carefully assess and identify their information and service needs. Consequently, better targeted information could be provided to caregivers through credible online sources. In this regard, an early assessment of caregivers' needs and digital skills demonstrates that large-scale actions aiming to equip informal caregivers with the digital skills they need to access digital support services are needed. This is key to enable informal caregivers to identify the available digital support services, and apply them to their own care situation. More research is therefore needed to examine the extent to which existing digital support services meet caregivers' information and service needs.

Addressing socioeconomic inequalities is likely to be key to reduce the digital divide in caregivers' use of the internet to access digital support services. As for the influence of age and education on the digital divide, healthcare professionals, service providers and social workers should pay particular attention to those caregivers who are older and less educated. Access to computers and internet connections at public facilities, such as local libraries, community centres and senior centres should be provided with extra support to accommodate caregivers' information needs and overcome any barriers of use.[15 19] Moreover, seminars and campaigns on how to access digital support services could enhance caregivers' digital skills and experiences. Tailor-made campaigns and classes for older and less-educated adults are needed to help address any barriers related to their use of computers and new technology.

Poor connectivity to the internet, particularly for informal caregivers in rural areas, is an obstacle to the use of any support service delivered over the internet. Policymakers should allocate funding for improving digital infrastructures in order to facilitate the deployment of digital support services and improve informal caregivers' access to these services. In this regard, an identification of sustainable business models, exchange of good practices, collection of evidence and a transferability of optimal solutions among localities, regions and countries are all important to continue allocating public funding for initiatives. Moreover, informal caregivers have concerns on data ownership and privacy of the data. Privacy concerns may be especially relevant to older informal caregivers, who voice the most concerns over the privacy and security of their information online. Digital support services should be sensitive to the privacy concerns of informal caregivers and the extent to which a technology might undermine their autonomy, control and dignity. In this context, blending online support with involving healthcare professionals in the provision of professional support leads to overcoming possible skepticism.

The finding that caregivers who indicate higher-intensity levels of caregiving are likely to engage in frequent internet use to access digital support services may suggest that the internet could be used to reach out to these caregivers and meet their information and

service needs. Online training materials, support groups, social networking systems for peer support and volunteer call networks could be used to reach out to caregivers.[62] Research is needed to further examine the effectiveness of digital support services in helping caregivers, if we are to improve these services and tailor them to the lives of those with substantial and unpredictable caring responsibilities.

## CONCLUSIONS

The findings from this study can provide guidance and assistance for the deployment of digital support services for informal caregivers. Nevertheless, due to rapid technological innovation, especially in this sector, continuous research needs to be conducted and guidelines for developing digital support services should be made adaptable to ongoing and future changes. The care sector is undergoing a fast transformation and expansion also due to the direct and indirect effects of the COVID-19 pandemic. Health and social care delivery systems experience a technologically supported transition towards home care. New technologies are being developed for informal caregivers and these tools may well offer benefits to many of them. It is widely acknowledged that caregivers are a group with high levels of unmet needs when it comes to their access to information and other services. Digital support services could be important tools to empower and support informal caregivers. On the other hand, it also needs to be recognized that informal caregivers are a diverse population, living in a wide range of personal and social circumstances. When it comes to policy and practice in relation to caregivers, similarly to other broad vulnerable groups, there is no 'one-size-fits-all' approach, and it is therefore important to consider the specific characteristics and needs of both caregivers and care recipients. Policy makers, healthcare professionals and all parties with an interest in supporting informal caregivers are encouraged to identify the outcomes that the latter regard as helpful, and to identify the interventions that can achieve such outcomes in consultation with them. This applies as much to the approach taken in relation to the development of digital support services as it does to other services. While digital support services have the potential to meet some of the needs of the caregivers, they cannot be seen as the only way to deliver information and support. These services represent only one of many instrument in a toolbox, and should therefore be tailored in a coordinated way with other existing services, such as respite care, access to training, and recognition of skills and work–life balance measures.

**Acknowledgements** The authors gratefully acknowledge the advice of the following experts: Marco Cucculelli, Micol Bronzini, Elizabeth Hanson, Lennart Magnusson and Stecy Yghemonos. The authors gratefully thank Sabrina Quattrini (Center for Socio-Economic Research on Aging of the Italian National Institute of Health and Science on Aging), Maria Blad (the Swedish Family Care Competence Center) and Olivier Jacqmain (Eurocarers Association) for their help in disseminating the survey questionnaire. The authors would also like to sincerely thank all the Italian and the Swedish care organizations who helped in disseminating the survey questionnaire.

**Contributors** Alhassan Yosri Ibrahim Hassan developed the research idea, designed the study, wrote the manuscript and is responsible for the overall content as the guarantor. Giovanni Lamura and Mariët Hagedoorn were involved in the conceptualization of the project and provided critical evaluation and approval of the final submitted manuscript.

**Funding** This research was funded by the European Union's Horizon 2020 research and innovation program under the Marie Skłodowska-Curie grant agreement number 814072 for the 4-year innovative training network ENTWINE informal care. This research was partially supported by Ricerca Corrente funding from the Italian Ministry of Health to IRCCS-INRCA.

**Competing interests** None declared.

**Patient and public involvement** Patients and/or the public were not involved in the design, or conduct, or reporting, or dissemination plans of this research.

**Patient consent for publication** Not applicable.

**Ethics approval** Permission to conduct the study was granted by the ethics committee of the faculty of economics, Marche Polytechnic University and was approved by the executive board on 2 November 2020 (1026353).

**Provenance and peer review** Not commissioned; externally peer reviewed.

**Data availability statement** No data are available.

**ORCID iD**
Alhassan Yosri Ibrahim Hassan http://orcid.org/0000-0003-3375-061X

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
