## [Reviewer comments · BMJ Open]

ARTICLE DETAILS

TITLE (PROVISIONAL)	Predictors of Digital Support Services Use by Informal Caregivers: A Cross-Sectional Comparative Survey
AUTHORS	Hassan, Alhassan Yosri Ibrahim; Lamura, Giovanni; Hagedoorn, Mariët

VERSION 1 – REVIEW

REVIEWER	Pendergrass, Anna Friedrich-Alexander-University Erlangen-Nuremberg
REVIEW RETURNED	10-Jan-2022

GENERAL COMMENTS	The manuscript has been reviewed following the STROBE guidelines. The protocol is scientifically sound and detailed. The research question is very important; especially during these times of the COVID 19 pandemic. There are a few components, which require more detail. The detail review is as follows and few suggestions are recommended: Title: Reading the title I would have not expected the comparison of predictors of digital support services in two different countries. The authors should consider mentioning this. Abstract: The abstract is clear and informative. Introduction: It includes the scientific background. However, I would recommend stating all objectives/aims at the end of the introductions. Now you can find them in paragraph 2 and 3. Method: Study design: I found lines 16-22 (page 6) a little confusing. For example, why would you mention a Dutch institution here while only using the data of Italian and Swedish institutions? Survey administration: Could you give examples for the different communication channels? Variables and measurement Since the variable “caregivers’ frequent use of the internet” is the dependent variable it would be interesting to know how exactly the answer categories of this variable were or what exactly frequent means. Did you have any missing data? If yes, how did you address them? Results: Tables: The tables are not in APA style. However, this is maybe not a requirement of the journal. Anyhow, the formatting of all tables is not consistent (e.g. compare page 11, lines 10-12 vs. 15-21). Please also look at table 2. The numbers are not in line with the variables.
--

	Discussion Research ethics approval was not mentioned.
--	---

REVIEWER	Najafi, Bijan Baylor College of Medicine
REVIEW RETURNED	23-Feb-2022

GENERAL COMMENTS	This study sought to examine the impact of Informal caregiver's characteristics including demographics, socioeconomic resources, and caregiving contact on the acceptability of using digital support services. To achieve this goal, authors collected on online survey of 663 individuals from two European countries: Italy and Sweden. Their results releveled several important factors influencing the use of digital support such as educational level, hours per week spent caring, gender, and few of socioeconomics resources. Overall this is a well written manuscript with potential impact in the field. Following are summary my major concerns and few comments that may assist in improving the manuscript:  • By a quick search through literatures, I found few publications which seem to explore similar questions in other nations (e.g., Shaffer et al, Informal Caregivers' Use of Internet-Based Health Resources, 2018; Kim, 2020, Caregivers' Information Overload and Their Personal Health Literacy; Ghahramani et al, 2021 Intention to Adopt mHealth Apps Among Informal Caregivers: Cross-Sectional Study; etc). What distinguishes this study compared to prior studies? If the contribution of this study is to discuss the acceptability of using digital resources in Europe, then authors want to explore whether their results are comparable with other countries like the comparable data available from USA • Further details is needed to describe the strategy of authors to minimize the potential bias. For instance, how many people were approached and how many returned the survey, what was the strategy to remind/follow-up for returning survey, what was the demographics of those who didn't return survey (if they are available) and whether these demographics are different compared to the cohort who returned the survey? • Considering that the survey done via online questionnaire, already the study is narrowed down to those who have internet access. This seems to be a major limitation in this study biasing toward users of digital resources. If this is the only source of recruiting participant, then it seems the abstract and inclusion criteria should specify that the cohort is limited to those who have access to internet. • The current inclusion and exclusion criteria seem to be too broad. For example I assumed that those with cognitive impairment, depression, and those with severe visual and hearing problems that may affect the ability to answer questionnaire were excluded? • Living in rural or in urban area and access to high speed internet may be also important factors affecting the perceive ease of use and perceived benefit of digital resources, none seems to be discussed in this manuscript. • There is lack of discussion on privacy concerns and data sharing concerns. These two factors could be major factors affecting the acceptability of using digital resources. • Another potential weakness is lack of an open ended question to collect some of the concerns of caregivers that may limit their ability in using digital resources. For instance, it may possible that
---

	there is not sufficient digital resources for caregiving tasks in their own native language.  • The results are not clear what are the major difference between subsample in Italy and Sweden (if there is any major difference) after adjusting by other co-variants like the level of education, age, gender, etc?
--	--

VERSION 1 – AUTHOR RESPONSE

Reviewer: 1

Dr. Anna Pendergrass, Friedrich-Alexander-University Erlangen-Nuremberg

Comments to the Author:

The manuscript has been reviewed following the STROBE guidelines. The protocol is scientifically sound and detailed. The research question is very important; especially during these times of the COVID 19 pandemic. There are a few components, which require more detail.

> Many thanks for your feedback and for your useful suggestions.

The detail review is as follows and few suggestions are recommended:

Title: Reading the title I would have not expected the comparison of predictors of digital support services in two different countries. The authors should consider mentioning this.

>The title has been updated according to the reviewer's feedback.

Abstract: The abstract is clear and informative.

Introduction: It includes the scientific background. However, I would recommend stating all objectives/aims at the end of the introductions. Now you can find them in paragraph 2 and 3.

>The introduction section has been rearranged according to the reviewer's feedback.

Method:

Study design: I found lines 16-22 (page 6) a little confusing. For example, why would you mention a Dutch institution here while only using the data of Italian and Swedish institutions?

>The University Medical Center Groningen (Netherlands) is the coordinator of the project. We thought to give information to the reader on the wider context where the data were collected. Nevertheless, based on the reviewer's feedback it was removed in order to avoid any confusion for the reader.

Survey administration: Could you give examples for the different communication channels?

>Definitely. The text has been updated with examples.

Since the variable "caregivers' frequent use of the internet" is the dependent variable it would be interesting to know how exactly the answer categories of this variable were or what exactly frequent means.

>In the "Methods" section, "Variables and Measurement" the definition of frequent has been mentioned as "those using the Internet at least several times per month to access digital support services were classified as "frequent users", while those accessing it less often were classified as "infrequent users".

Did you have any missing data? If yes, how did you address them?

>We only included completed responses

Results:

Tables:

The tables are not in APA style. However, this is maybe not a requirement of the journal. Anyhow, the formatting of all tables is not consistent (e.g. compare page 11, lines 10-12 vs. 15-21).

Please also look at table 2. The numbers are not in line with the variables.

>The tables' formatting has been updated.

Discussion

Research ethics approval was not mentioned.

>The “research ethics approval” was updated in the “Methods” section based on the editor’s feedback.

Reviewer: 2

Dr. Bijan Najafi, Baylor College of Medicine

Comments to the Author:

This study sought to examine the impact of Informal caregiver’s characteristics including demographics, socioeconomic resources, and caregiving contact on the acceptability of using digital support services. To achieve this goal, authors collected an online survey of 663 individuals from two European countries: Italy and Sweden. Their results revealed several important factors influencing the use of digital support such as educational level, hours per week spent caring, gender, and few of socioeconomic resources. Overall this is a well written manuscript with potential impact in the field. Following are summary of my major concerns and few comments that may assist in improving the manuscript:

>Many thanks for your feedback and for your useful suggestions.

- By a quick search through literatures, I found few publications which seem to explore similar questions in other nations (e.g., Shaffer et al, Informal Caregivers’ Use of Internet-Based Health Resources, 2018; Kim, 2020, Caregivers’ Information Overload and Their Personal Health Literacy; Ghahramani et al, 2021 Intention to Adopt mHealth Apps Among Informal Caregivers: Cross-Sectional Study; etc). What distinguishes this study compared to prior studies? If the contribution of this study is to discuss the acceptability of using digital resources in Europe, then authors want to explore whether their results are comparable with other countries like the comparable data available from USA

>Indeed, we have included these literatures in our introduction and discussion. We found that our results are consistent with the literature. Based on the reviewer’s feedback we updated the “Discussion” section.

- Further details is needed to describe the strategy of authors to minimize the potential bias. For instance, how many people were approached and how many returned the survey, what was the strategy to remind/follow-up for returning survey, what was the demographics of those who didn’t return survey (if they are available) and whether these demographics are different compared to the cohort who returned the survey?

>We provided more details in the “Methods” section as per the reviewer’s feedback. Nevertheless, there are some details missing such as the demographics of those who did not return the survey. We acknowledge the limitation in the “Discussion” section under “limitations”

- Considering that the survey done via online questionnaire, already the study is narrowed down to those who have internet access. This seems to be a major limitation in this study biasing toward users of digital resources. If this is the only source of recruiting participant, then it seems the abstract and inclusion criteria should specify that the cohort is limited to those who have access to internet.

>We agree with the reviewer that this is a major limitation in this study as with other online studies. This is mentioned in the “Discussion” section under “limitations”. As per the reviewer’s feedback, we updated both the abstract and the inclusion criteria to reflect this fact.

- The current inclusion and exclusion criteria seem to be too broad. For example I assumed that those with cognitive impairment, depression, and those with severe visual and hearing problems that may affect the ability to answer questionnaire were excluded?

>The inclusion and exclusion criteria have been updated according to the reviewer’s feedback.

- Living in rural or in urban area and access to high speed internet may be also important factors affecting the perceived ease of use and perceived benefit of digital resources, none seems to be discussed in this manuscript.

- >The “Discussion” section has been updated according to the reviewer’s section.
- There is lack of discussion on privacy concerns and data sharing concerns. These two factors could be major factors affecting the acceptability of using digital resources.
- >The “Discussion” section has been updated according to the reviewer’s section.
- Another potential weakness is lack of an open ended question to collect some of the concerns of caregivers that may limit their ability in using digital resources. For instance, it may possible that there is not sufficient digital resources for caregiving tasks in their own native language.
- >We added this limitation to the “limitations” under “Discussion”
- The results are not clear what are the major difference between subsample in Italy and Sweden (if there is any major difference) after adjusting by other co-variants like the level of education, age, gender, etc?
- >There were no major differences. We report on that in the “Results” and “Discussion” sections.

VERSION 2 – REVIEW

REVIEWER	Najafi, Bijan Baylor College of Medicine
REVIEW RETURNED	29-Mar-2022
GENERAL COMMENTS	Authors have answered all my major concerns and I believe the manuscript has sufficient scientific merit to be accepted in BMJ.